# Clinical Theranostics in Recurrent Gliomas: A Review

**DOI:** 10.3390/cancers16091715

**Published:** 2024-04-28

**Authors:** Austin R. Hoggarth, Sankar Muthukumar, Steven M. Thomas, James Crowley, Jackson Kiser, Mark R. Witcher

**Affiliations:** 1Department of Neurosurgery, Carilion Clinic, 1906 Belleview Avenue, Roanoke, VA 24014, USA; mrwitcher@carilionclinic.org; 2Virginia Tech Carilion School of Medicine, 2 Riverside Circle, Roanoke, VA 24016, USA; smuthuk24@vt.edu (S.M.); smthomas1@carilionclinic.org (S.M.T.); 3School of Neuroscience, Virginia Polytechnic Institute and State University, Blacksburg, VA 24061, USA; 4Carilion Clinic Radiology, Roanoke, VA 24016, USA; jrcrowley@carilionclinic.org (J.C.); jwkiser@carilionclinic.org (J.K.)

**Keywords:** glioblastoma, GBM, theranostics, functional quality of life, QOL

## Abstract

**Simple Summary:**

This paper discusses the challenges in treating high-grade gliomas, particularly glioblastomas, which are aggressive brain tumors with high recurrence rates. It highlights the limitations of current diagnostic imaging modalities, such as gadolinium-based MRI, in accurately detecting tumor recurrence versus treatment-related changes. The paper explores the emerging role of positron emission tomography (PET) in glioma imaging, especially with the use of radiotracers like PSMA, which can help differentiate between tumor recurrence and treatment effects. Furthermore, the paper reviews the concept of theranostics, which integrates diagnostics and therapy, offering a targeted approach to glioma treatment. It discusses various radioligands, including PSMA, 213Bi-DOTA-substance P, 90Y-DOTATOC, ^18^F-FDOPA, p-[131I]-iodo-L-phenylalanine, and ^18^F-GE-180, which have shown promise in diagnosing and treating recurrent gliomas. The potential of theranostics to minimize systemic toxicity and improve treatment outcomes is emphasized. Moreover, the paper highlights the importance of considering quality-of-life (QOL) outcomes in glioma patients, as conventional treatments often have significant impacts on patients’ well-being. While theranostics offers a personalized approach to treatment, its potential promise for functional outcomes and QOL needs further investigation. The paper suggests that future research should focus on understanding the broader implications of theranostics on patient well-being, incorporating factors such as demographics, tumor characteristics, and treatment-related effects into QOL assessments. Overall, the paper underscores the potential of theranostics to revolutionize glioma management and improve patient outcomes in the future.

**Abstract:**

Gliomas represent the most commonly occurring tumors in the central nervous system and account for approximately 80% of all malignant primary brain tumors. With a high malignancy and recurrence risk, the prognosis of high-grade gliomas is poor, with a mean survival time of 12–18 months. While contrast-enhanced MRI serves as the standard diagnostic imaging modality for gliomas, it faces limitations in the evaluation of recurrent gliomas, failing to distinguish between treatment-related changes and tumor progression, and offers no direct therapeutic options. Recent advances in imaging modalities have attempted to address some of these limitations, including positron emission tomography (PET), which has demonstrated success in delineating tumor margins and guiding the treatment of recurrent gliomas. Additionally, with the advent of theranostics in nuclear medicine, PET tracers, when combined with therapeutic agents, have also evolved beyond a purely diagnostic modality, serving both diagnostic and therapeutic roles. This review will discuss the growing involvement of theranostics in diagnosing and treating recurrent gliomas and address the associated impact on quality of life and functional recovery.

## 1. Introduction

Gliomas, or tumors arising from glial cells, are the most commonly occurring primary brain tumors. High-grade gliomas represent the most devastating form of gliomas, rendering patients with a median survival time of just 12–18 months after undergoing a comprehensive treatment regimen of maximal surgery, chemotherapy, and radiation therapy [1]. Further complicating the treatment of these malignant neoplasms is the extremely high recurrence rate, at around 90% within 2 years of the initial diagnosis [2]. Furthermore, these tumors are estimated to recur within a median time of 6.7 months, highlighting the need for serial imaging and management. A summary of PET radiotracers used in glioma imaging is provided in Table 1 [3,4,5,6,7,8,9,10,11,12,13].

Gadolinium-based contrast-enhanced MRI (CE-MRI) serves clinically as the first-line diagnostic tool for imaging gliomas. Damage to the blood–brain barrier (BBB) generally allows gadolinium dye to accumulate in areas of tumor growth, showing hyperintensities in areas where the tumor is located. While it effectively localizes primary tumors, MRI faces problems with diagnosing recurrent gliomas, often failing to distinguish between tumor recurrence and treatment-related changes such as radiation necrosis, as both may involve BBB disruption [14]. One study found that CE-MRI exhibited relatively low sensitivity (68%) and specificity (77%) in detecting recurrent high-grade gliomas [15]. Addressing some of these limitations of imaging high-grade gliomas is essential: unnecessarily treating patients without tumor recurrence may lead to unnecessary surgeries and extended therapy, whereas failure to treat recurrent gliomas may rapidly worsen the prognosis of patients with this condition.

Positron emission tomography (PET) is another imaging modality that has been used to diagnose gliomas. PET imaging utilizes labeled radiotracers to evaluate metabolic activity in the body and is particularly useful in the realm of oncology; cancer cells exhibit higher levels of metabolic activity and thus exhibit higher levels of uptake on PET imaging. In the realm of gliomas, PET has demonstrated utility in delineating tumor extent, guiding treatment planning, and assessing post-treatment outcomes [16]. One of the key advantages that PET presents is its ability to distinguish between radiation necrosis and tumor recurrence, exhibiting higher uptake levels of the radiotracer when imaging gliomas.

While PET is rapidly growing as an effective diagnostic tool, the same biomarkers that are used for imaging can also be used to help guide treatment in patients with cancer. Theranostics is a novel medical term that refers to a combination of diagnostics and therapy. In practical use, theranostics utilizes a combination of targeting molecules paired with either diagnostic or therapeutic radionuclides [17]. This allows for a more targeted approach in managing various forms of cancers, to which the targeting ligands ultimately bind specifically. While in use primarily for other forms of cancer to this point, theranostics may serve a valuable role in treating patients with gliomas. This review will explore the involvement of theranostics in recurrent gliomas.

## 2. Theranostics Overview

Integrating diagnosis with the treatment of gliomas through theranostics may have a role in improving the prognosis of patients suffering from this condition. While the current treatment regimen for gliomas is well established, patients often suffer from systemic toxicity after undergoing chemotherapy and experience debilitating side effects [18,19,20]. Theranostics potentially allows for a more targeted approach to treatment, with the localization of tumor tissue using imaging followed by the destruction of these same tissues with high-dose radiation [21], thus mitigating the systemic adverse effects that patients face.

Originally developed for use in imaging and treating thyroid cancer using radioactive iodine, theranostics has since been utilized in other forms of cancer, including paragangliomas and neuroendocrine tumors [21]. PET, SPECT, and nuclear scintigraphy are the most well-established imaging tools in theranostics, allowing for the imaging of gamma rays emitted by injected radioactive particles. From a therapeutic aspect, there are two primary forms of particle radiation used: alpha and beta. Alpha particles deliver high amounts of radiation over a small tissue volume and allow for a more targeted approach than beta particles, which penetrate more into the surrounding tissue [22].

## 3. Theranostic Use in Gliomas

PET has emerged as the currently most effective imaging modality in theranostics, owing to the relative ease of using the same target molecule (probe) to image and treat cancers [23]. One probe, prostate-specific membrane antigen (PSMA), has been increasingly used in PET imaging in recent years and has since emerged as a useful target for theranostics. In addition to exhibiting overexpression in prostate cancer, this antigen has also been identified in the highly vascularized endothelium of other tumors, most notably glioblastomas. Recent studies have demonstrated the effectiveness of ^68^Ga-PSMA PET imaging of gliomas due to the high tumor-to-brain ratio (TBR) of this radiotracer [24]. One theranostic partner for ^68^Ga-PSMA is ^177^Lu-PSMA-617, a beta-emitting compound containing lutetium that similarly uses PSMA as its target molecule, delivering radiation to the PSMA-expressing tumor cells [25]. ^177^Lu-PSMA-617 has proven to be a valuable theranostic tool in the realm of recurrent gliomas. One study showed a clinically significant reduction in ^68^Ga-PSMA on PET/CT imaging after several cycles of ^177^Lu-PSMA-617 therapy, also resulting in symptomatic improvement [25]. Another study performed ^177^Lu-PSMA-617 treatment in a patient with a recurrent GBM after initial imaging with ^68^Ga-PSMA-11 PET/CT and found that the tumor exhibited decreased uptake of ^68^Ga-PSMA-11 PET/CT after 14 days of treatment [26]. However, while the initial results of ^177^Lu-PSMA-617 have been promising, clinicians still face challenges in implementing this radioligand in practice. A study involving three patients treated with ^177^Lu-PSMA-617 found that the dose of radiation administered was too low for therapeutic effects, failing in the balance between delivering an effective tumor dose and causing harmful radiation effects [27].

## 4. Current Radioligands

In addition to PSMA, other theranostic ligands have been developed and studied for gliomas in recent years and have shown promising preliminary results. 213Bi-DOTA-substance P was shown in one study to effectively treat secondary GBM through alpha emission [28]. Substance P binds to neurokinin-1 (NK-1) receptors, which are highly expressed in all types of gliomas [29]. This study also demonstrated that the median overall survival in patients after treatment is similar to that in patients undergoing alternative treatment options.

In a separate study, ^90^Y-DOTATOC, a beta emitter, was used to treat patients with recurrent GBM. This ligand binds to the somatostatin receptor, which is upregulated in GBM16. These patients showed a partial to complete response to treatment, and patients reported improved quality of life following therapy [30].

One of the most well-studied radiotracers in clinical practice is ^11^C-methionine (^11^C-MET), which is taken up through L-amino acid transporters (LATs). While it exhibits an increased tumor-to-brain contrast ratio on imaging, it has a few limitations, including a short half-life of just 20 min and increased uptake in necrotic tissue secondary to radiation therapy [31]. As a result, many new tracers are using radioisotopes such as fluorine-18, which has a longer half-life at 110 min. Despite these shortcomings, ^11^C-MET has been shown to be effective in evaluating treatment response [32] and diagnosing glioma recurrence [33], suggesting its continued use in clinical practice.

^18^F-FDOPA has been previously shown to detect contrast-enhancing and nonenhancing brain tumors. Brain tumor uptake of ^18^F-FDOPA is predominantly dependent on LAT expression and activity [34]. Hermann et al. identified that ^18^F-FDOPA PET had a diagnostic accuracy of 82% (sensitivity, 89.6%; specificity, 72.4%) in differentiating recurrent GBM from treatment-associated changes and that a mean lesion-to-normal brain tissue (L/NB) ratio of 1.8 best discriminated progression-free survival (39.4 months if mean L/NB <1.8 and 9.3 months if mean L/NB ratio was ≥1.8) [35]. Visual (5-point scale) and semiquantitative indices (i.e., lesion-to-striatum ratio and L/NB ratio) of ^18^F-FDOPA PET imaging yielded similar detection accuracies of GBM recurrence (82% and range of 77–82%, respectively); however, none of the aforementioned indices were able to predict overall survival [35].

^18^F-FET is another radiotracer that has been used in the setting of high-grade gliomas, particularly in Europe. Like ^18^F-FDOPA, ^18^F-FET is taken up through L-amino acid transporters. It has demonstrated remarkable diagnostic accuracy (96%) in differentiating treatment-related changes from tumor recurrence. In the same study, the TBR of ^18^F-FET also served as an effective prognosticator of overall survival; patients who had a TBR_max_ < 2.3 were found to live significantly longer (23 months in comparison to 12 months) [36]. This tracer has also been used to evaluate treatment response, with one study noting a significantly longer median disease-free survival (10 months vs. 3 months) in patients who demonstrated an early response to radiochemotherapy, as determined by a decrease in ^18^F-FET uptake [37]. It is worth noting that this radiotracer recently obtained an expanded access designation through the Food and Drug Administration as an investigational new drug, with the hope of expanding its use in glioma patients in the United States.

Another radioligand associated with the L-amino acid transporter system is p-[(131)I]-iodo-L-phenylalanine (131IPA). 4-Iodo-L-phenylalanine is a derivative of the amino acid L-phenylalanine, and Iodine-131 serves as a beta emitter. A study of five patients with recurrent GBM who underwent 131IPA and hypo-fractionated external beam radiation therapy demonstrated that the treatment was well tolerated and safe but less effective than suggested by prior animal studies [38].

Apart from amino acid transport, other tracers have been developed that function under different mechanisms of action. ^18^F-FMISO is a tracer that works under hypoxic conditions; high-grade gliomas are associated with a greater degree of hypoxia, and thus, the uptake of ^18^F-FMISO will be greater in these tumors [39]. In addition to localizing areas of hypoxia within the brain, this tracer has been proven to be an effective marker of treatment response to anti-VEGF therapy, as Barajas et al. noted a marked decrease in ^18^F-FMISO uptake in patients after undergoing bevacizumab therapy [40].

A well-studied radioligand is ^18^F-GE-180, a receptor for the mitochondrial translocator protein (TSPO). TSPO is overexpressed in activated microglia and macrophages, and its expression has been reported to be upregulated in gliomas [41,42]. Albert et al. investigated TSPO PET in four patients with recurrent isocitrate dehydrogenase wildtype high-grade gliomas and found the median maximal tumor-to-background ratio to be 5.86 [43]. The high tumor-to-background ratio could be attributed to very low ^18^F-GE-180 binding to normal brain tissue (median background uptake 0.47; range 0.37–0.93) [43]. One study found no significant parameter differences for either ^18^F-GE-180 and ^18^F-FET PET when comparing newly diagnosed and recurrent gliomas, highlighting that ^18^F-GE-180 uptake is not specific to only recurrent cases of glioma [44]. A study investigated 25 patients with recurrent glioma, and of these 25 patients, 3 had histologically verified malignant transformation, and 3 of these patients showed no malignant transformation (histological verification) at the time of ^18^F-GE-180 PET [45]. Patients with malignant transformation demonstrated high localized uptake of ^18^F-GE-180 (tumor-to-background ratios of 5,36, 7,64, and 6,30), whereas patients without malignant transformation did not show detectable uptake of ^18^F-GE-180 upon visual analysis [45]. Quach et al. correlated TSPO PET signal using ^18^F-GE-180 with clinical outcomes in a cohort of 88 patients with recurrent glioma [34]. They found that TSPO tracer uptake significantly correlated with tumor grade at recurrence and that a median maximum standardized uptake value < 1.68 predicted significantly longer median post-recurrence survival (41.6 vs. 12.6 months) and median time to treatment failure (14.9 vs. 6.2 months) [34].

Another novel radioligand that has been developed recently is FAPI (fibroblast activator protein inhibitor). Fibroblast activator proteins (FAPs) are overexpressed in cancer cells; thus, ligands that bind to these proteins serve as effective targets for PET imaging. One study found that the uptake of FAPI was greater in high-grade gliomas compared to normal brain tissue, indicating its potential use in glioma imaging. This tracer has also been shown to identify gliomas that are poorly delineated on MRI imaging, likely owing to its high tumor-to-brain contrast ratio [46].

TLX250-CDx is a new tracer that targets carbonic anhydrase (CAIX), an enzyme that is overexpressed in many cancer cell lines in response to hypoxia [47]. Originally developed for renal cancer, this radiotracer has indications for use in gliomas, which also overexpress this enzyme. While it has not yet been fully investigated in patients with gliomas, it remains a promising target, particularly as tumors that overexpress CAIX portend a worse prognosis and tend to be resistant to immunotherapies. A summary of current treatments for recurrent glioma is provided in Table 2.

## 5. Delivery of Radiotracers

Using innovative approaches beyond intravenous administration is needed to achieve optimal radiotracer dose absorption [48]. With respect to modifying the administration technique, Vonken et al. evaluated the intravenous administration of ^177^Lu-HA-DOTATATE followed by intraarterial treatment cycles (median number of cycles: two) in four patients with surgery- and radiotherapy-refractory meningiomas [49]. Both planar and SPECT/CT imaging showed increased tracer accumulation in the target lesion after intraarterial radiotracer administration cycles relative to intravenous administration only, with average increases of 220% and 398%, respectively. No unexpected adverse events occurred [49]. Future studies should explore the dosimetric analysis of intraarterial radiotracer administrations over multiple timepoints in patients with recurrent glioma. Another novel administration of radionuclides is radioembolization [50]. Pasciak et al. demonstrated the feasibility and safety of yttrium-90 radioembolization in a canine model of brain cancer [51]. The use of radioembolization administration of radiotracers should be considered in future studies on recurrent glioma.

An additional approach is the combination of radiotracer therapy with techniques that penetrate the blood–brain barrier and the blood–tumor barrier, such as convection-enhanced delivery, focused ultrasound, and the direct delivery of the radiotracer to the glioma resection cavity [52]. Beyond the aforementioned interventional approaches, techniques such as the chemical modification of existing radiotracers to optimize tracer half-life [53] and the use of a monoclonal antibody-based probe and a nanoparticle-based probe [53,54] to target molecules in glioma have the potential to improve the delivery of radiotracer therapy. However, the aforementioned approaches have been primarily investigated in preclinical models, and there is a great need for future studies to apply the multimodal delivery of radiotracers in recurrent glioma patients. Prior studies have shown an improvement in the radiotracer dose by combining external beam radiation therapy with radiotracer therapy in the treatment of advanced meningioma. This improvement is due to the difference in radiation fields between the two techniques, which can lead to more localized radiation. As re-irradiation is a treatment for localized recurrent glioma [55], future studies should explore combined external beam radiation and radiotracer therapy in recurrent glioma patients.

## 6. Radiotracer Dosimetry

A technical limitation hindering the increased use of theranostics in recurrent glioma care is the optimization of the PET radiotracer dosage to balance radiation exposure with an effective tumor dose. The optimization of the dose of an intravenously injected radiotracer aligns with the “as low as reasonably achievable” principle and can lead to the cost-effective use of PET imaging resources. Transitioning from multiple radiotracer doses to personalized tumor dosimetry can advance theranostic use in the recurrent glioma patient population. The medical internal radiation dose formalism is typically used for radiotracer dosimetry calculations for organs at risk and target organs [56]. Alternative radiotracer dosimetry methodologies include three-dimensional image-based dosimetry and Monte Carlo simulations. Beyond alpha and beta emission, auger electrons have a short range of activity (<100 nm), inducing DNA damage and cell death [57,58]. Advances in dosimetry approaches for radiotracer therapy, including cellular dosimetry and microdosimetry, are currently being explored at the preclinical level [58]. Innovative dosimetry approaches and the use of auger electrons represent a new avenue of theranostics research on recurrent gliomas.

Karakatsanis et al. provide an approach to optimize the radiotracer dose based on a noise equivalent count rate (NECR)–dosage curve based on simulations and phantom experiments that consider patient factors (e.g., patient attenuation volume) and scanner system factors (e.g., scanner energy, coincidence time window) [59]. Given the heterogeneous nature of recurrent gliomas, using a constant NECR score as a dosage optimization criterion may be needed to standardize the high signal-to-noise ratio for PET data. Karakatsanis et al. suggest that an optimal constant NECR score could be the maximum predicted NECR for the most obese patients of a representative patient population [59]. Lower NECR scores can be compensated by increasing the PET scan duration, particularly in PET/MRI protocols [59]. Therefore, a consensus regarding an acceptable NECR score for radiotracers in recurrent glioma management is needed. Additionally, standardizing the patient and scanner system factors assessed in simulations and phantom experiments can improve the comparison of radiotracers, as individual recurrent glioma patient and PET scanner system factors can be substituted accordingly. Future studies should provide data on radiotracer dose calculation based on constructed NECR–dosage curves. Furthermore, a reduction in the required radiotracer dose will occur when advances in PET technology, such as long-axial field-of-view PET, are applied to recurrent glioma patients [60].

## 7. Functional Neuro-Oncologic Implications

Quality-of-life (QOL) measurement has become increasingly important as an outcome metric in brain cancer patients [61]. QOL, a multifaceted concept, encompasses an individual’s overall well-being and satisfaction with life, spanning various dimensions, such as physical and functional status, emotional well-being, and social well-being [62]. High-grade glioma patients encounter numerous challenges related to QOL, including general symptoms like headaches, anorexia, nausea, seizures, and insomnia, as well as symptoms stemming from neurological deterioration, such as motor deficits, personality changes, cognitive deficits, aphasia, or visual field defects [63,64]. The use of theranostic treatment options, whether in conjunction with or instead of current standard therapies, has the potential to lessen the burden experienced by patients treated for high-grade glioma.

Glioblastoma typically necessitates a multimodal treatment approach. In general, the initial management involves surgical resection to maximal safe margins, aiming to minimize the burden of cancerous tissue while preserving essential brain function. Following surgery, a combination of radiation therapy and chemotherapy is often employed to target residual cancer cells [65].

Radiation-induced cognitive impairment has been linked to declines in verbal memory, spatial memory, attention, and problem-solving skills [18]. It is hypothesized that these late effects, including cognitive impairment, result from complex interactions among various brain cell types, including astrocytes, endothelial cells, microglia, neurons, and oligodendrocytes [66,67,68]. Traditionally, alterations in vascular and neuroinflammatory glial cell populations induced by radiation were thought to be responsible for radiation-induced brain injury [69]. Preclinical studies have particularly focused on the hippocampus, a critical region for learning and memory, where radiation appears to inhibit neurogenesis, disrupt neuronal function, and trigger neuroinflammation [69].

In long-term survivors treated with radiation therapy, cognitive function was notably more impaired in those who received whole-brain radiation compared to focused radiotherapy [68]. However, it is often challenging to distinguish between the effects of treatment- and tumor-related factors on cognitive decline. Some studies suggest that patients with tumors may experience poor quality of life regardless of whether radiation was administered [19]. Moreover, modern radiotherapy techniques may not induce the same long-term cognitive effects as whole-brain radiotherapy [70].

The assessment of chemotherapy-related impacts on quality of life (QOL) faces challenges in distinguishing between the effects of chemotherapy itself and those of other treatments or the tumor [61]. QOL among newly diagnosed glioblastoma multiforme (GBM) patients undergoing either radiotherapy alone or radiotherapy with concurrent and adjuvant temozolomide (TMZ) has been examined [71]. Both treatment groups exhibited substantial impairment compared to historical controls, with no significant overall decrease in QOL observed throughout treatment. Patients receiving TMZ experienced heightened levels of vomiting, anorexia, constipation, and decreased social functioning, with increased fatigue during radiation therapy [64]. It is unsurprising that patients who responded to TMZ reported improvements across multiple QOL domains [64].

The concurrent use of medications can complicate the study of therapy-related symptoms. Common medications administered to brain tumor patients, such as seizure medications and steroids, can negatively impact physical, emotional, and cognitive functioning [20]. Specifically, anti-epileptic medications have been associated with cognitive dysfunction, while corticosteroids have been linked to depression in high-grade glioma patients and reduced survival in recurrent malignant glioma patients [69,72,73].

Currently, there are no studies evaluating the impact of glioma theranostics on functional QoL. As the field is showing promise in both diagnostic and therapeutic value, the use of these multimodal compounds may offer significant benefits to QOL. Its current potential use is in recurrent gliomas to help differentiate recurrent tumor vs. treatment effects. At recurrence, many of the factors playing a role in functional QoL will have impacted outcomes. Given the unique role of theranostics in the direct specific targeting of glial tumors, we anticipate that its use would likely minimize the systemic effects that often plague functional QoL outcomes associated with other therapeutic options.

As theranostic treatments are being further developed and studied for the treatment of gliomas, QOL studies should follow suit. Liu et al. (2009) proposed a model for future QOL research, outlining a method for analyzing the various factors contributing to a patient’s overall QOL. These factors encompass patient-related elements such as demographic characteristics and comorbidities, which can influence how patients perceive and experience symptoms [61]. Additionally, tumor-related factors such as laterality, size, and location play a role in shaping the specific neurological symptoms experienced by brain tumor patients. Finally, treatment-related factors, including surgery, radiation, chemotherapy, and concomitant medications, can either exacerbate or alleviate symptoms impacting QOL, including the functional aspects thereof [61].

## 8. Challenges and Limitations

While the field of theranostics has shown promise in delivering personalized care for patients, it has faced some intrinsic barriers that have prevented its implementation on a larger scale, potentially because personalized theranostic treatment may fall at odds with the standard of care at many institutions. Clinicians may also face challenges with determining patient eligibility for theranostic treatment, as clinical trials thus far have utilized highly specific inclusion and exclusion criteria that may not be truly representative of the patient population. Further complicating the widespread implementation of theranostics are logistical concerns, such as the cost and availability of medical cyclotrons and theranostic agents at therapy sites and the proper administration of the therapies by nuclear medicine specialists. Even with sufficient resources present at hand, the exact treatment that a patient will receive may be highly individualized, in terms of which radionuclide they receive and at what dose. Thus, the planning stages of glioma treatment with theranostics may be extensive, which may come into conflict with the desired standard of care. As a result, additional considerations need to be made in terms of integrating theranostics with other methods of cancer treatment, such as surgical intervention and chemotherapy.

The use of theranostics in gliomas in particular poses additional difficulties. Due to the relative heterogeneity of these tumors, particularly high-grade gliomas, developing tracers that fit an individual’s tumor profile is a challenging ordeal. As such, many of the current tracers available have a diagnostic component but not a therapeutic analog. Furthermore, although these theranostic molecules exert their effects on very selective regions to which they bind in the tumor tissue, they do emit some degree of radiation to healthy brain tissue, causing collateral damage and subsequent cytotoxic effects. Therefore, understanding the safety profile and dosimetry of these tracers is an ongoing process that needs to be addressed before theranostics becomes fully implemented in glioma care. Mitigating these various concerns may be a challenging endeavor; however, with recent advancements in imaging modalities such as PET/CT, the rapid development of novel radiotracers, and the growing experience of skilled nuclear medicine personnel, the role of theranostics will continue to evolve and impact clinical care.

## 9. Conclusions

Glioblastoma represents a formidable challenge in neuro-oncology, characterized by an aggressive nature, high recurrence rates, and limited treatment options. Current diagnostic modalities, such as gadolinium-based contrast-enhanced MRI, while useful in initial tumor localization, often fall short of distinguishing between tumor recurrence and treatment-related changes, necessitating the exploration of alternative imaging techniques. Positron emission tomography (PET), particularly using radiotracers like PSMA (prostate-specific membrane antigen), has shown promise in both accurately delineating tumor extent and guiding treatment decisions, including the use of theranostic agents. Theranostics, a burgeoning field that integrates diagnostics and therapy, offers a targeted approach to glioma treatment, potentially reducing or mitigating the systemic toxicity associated with conventional chemotherapy, as well as the possibility of improving patient outcomes. By employing radioligands like PSMA and others, theranostics not only aids in accurate tumor localization but also facilitates the targeted delivery of radiation therapy, offering a personalized approach to glioma management.

Moreover, as the field of theranostics continues to evolve, its impact on functional outcomes and quality of life (QOL) for glioma patients warrants further investigation. While the existing literature underscores the detrimental effects of conventional treatments on QOL, the potential of theranostics to minimize systemic adverse effects and improve treatment efficacy holds promise for enhancing patient well-being. Future research endeavors should focus not only on assessing the clinical efficacy of theranostic agents but also on understanding the broader implications for functional QOL outcomes. By incorporating patient demographics, tumor characteristics, and treatment-related factors into QOL assessments, researchers can elucidate the multifaceted influences on patient well-being, paving the way for more holistic and patient-centered glioma care in the era of theranostics.

## Figures and Tables

**Table 1 cancers-16-01715-t001:** Summary of PET radiotracers used in glioma imaging.

Tracer	Mechanism of Action	Advantages	Drawbacks
^18^F-FDG	Glucose metabolic activity	Widely available	High uptake in normal brain tissue
^11^C-MET	Amino acid transport	Effective for treatment planning and evaluation	Short half-life
^18^F-FET	Amino acid transport	Predictive of overall survival Helpful in evaluating tumor response	Limited availability
^18^F-FDOPA	Amino acid transport	Treatment planning and evaluation	Increased uptake in striatum Limited availability
^18^F-FMISO	Hypoxia marker	Can identify small tumor regions Treatment evaluation	Not yet validated for diagnostic performance
^18^F-GE-180	Neuroinflammation	Treatment evaluation	Not yet validated for diagnostic performance
^68^Ga-PSMA-11	Glioma neovasculature	Extremely high tumor-to-brain ratio Treatment evaluation	Not yet validated for diagnostic performance in gliomas

**Table 2 cancers-16-01715-t002:** Summary of current treatments for recurrent glioma.

Treatment Modality	Advantages	Limitations	Adverse Effects
Surgical resection	- Relief of symptomatic mass effect	- Applicable to only 20–30% of patients	- Surgical biopsy complications
	- Use of surgical adjuncts (e.g., 5-ALA-guided resection)	- Indications are not firmly established	- Wound healing complications associated with pretreatment with bevacizumab
	- Innovative approach (e.g., laser interstitial thermal therapy)		
Chemotherapy	- Bevacizumab improves quality of life with steroid-sparing effect	- No superior systemic agent	- Treatment toxicity
	- Genotype-targeted therapy	- Use of certain agents limited to clinical trial setting	
		- Poor response rate and overall survival	
Radiation Therapy	- Multiple radiation modalities	- More prospective data needed	- Treatment toxicity
	- Concurrent administration of bevacizumab	- Limited tumoral dose	- Late radiation necrosis

Abbreviation: 5-aminolevulinic acid, 5-ALA.

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
