# Peer review of "Clinical Theranostics in Recurrent Gliomas: A Review"

_cancers, 2024, doi:10.3390/cancers16091715_

Round 1

Reviewer 1 Report

Comments and Suggestions for Authors

The manuscript "Theranostics in Recurrent Gliomas: A review" discusses a recent and very intresting aspect of the diagnosis and management of brain gliomas. The integration of diagnostics and therapy offers a new targeted approach to glioma treatment, which improves the patient outcome. The article is well written n and the review is complete. Thus, I reccomend the publication.

Author Response

Minor revisions for reference numbering; instead of superscript- it is just [#] now. Also reworded highlighted sections that were deemed similar to prior publications and changed for better syntax.

Reviewer 2 Report

Comments and Suggestions for Authors

Comments:

This is a review regarding current theranostics (diagnosis and therapy) of recurrent gliomas. The review is fairly well-written. There were many typos in the TEXT.

  1. Page 2, line 52 – 53: the sentence “A comparison of current treatment modalities for recurrent gliomas is provided in Table 1” should be changed into “Summary of PET radiotracers used in glioma imaging is provided in Table 1”.

  1. Page 2, Table 1: “Metabolic activity” should be better changed into “glucose metabolic activity”.
  2. Page 4, line 126: the full name for the abbreviation “NK-1” should be given for the first time in the text.
  3. Page 4, line 138: “this tracer…” should be better changed into “11C-MET”.

  1. Page 5, line 210-211: the sentence “A summary of PET radiotracers used in glioma imaging is provided in Table 2” should be changed into

“Summary of Current Treatments for Recurrent Glioma is provided in Table 2”.

  1. Page 5, line 190: (….ratio of 5,36, 7,64 and 6,30) should be changed into (…. ratio of 5,36, 7,64 and 6,30, respectively)

  1. Page 5, last paragraph: The letter size of this paragraph is smaller than the other parts of text.  

  1. Page 6, line 221, the sentence “ 1220% and 1398% “ should be changed into “220% and 398%”

  1. Page 8, line 305: the word “my “ should be corrected to “may”.

  1. Page 8, line 130: the full name for the abbreviation “TMZ” should be given for the first time in the text.

  1.  The tracers such as 18F-FDG, 11C-MET and 18F-FET should be written in text as 18F-FDG, 11C-MET and 18F-FET.  The same for other tracers. The mass numbers of tracers should be superscripted.

  1. The References should be carefully checked according to the instructions of authors of “Cancers”.
Comments on the Quality of English Language

Only minor editing of English required 

Author Response

This is a review regarding current theranostics (diagnosis and therapy) of recurrent gliomas. The review is fairly well-written. There were many typos in the TEXT.

Page 2, line 52 – 53: the sentence “A comparison of current treatment modalities for recurrent gliomas is provided in Table 1” should be changed into “Summary of PET radiotracers used in glioma imaging is provided in Table 1”. Changed

Page 2, Table 1: “Metabolic activity” should be better changed into “glucose metabolic activity”. Changes

Page 4, line 126: the full name for the abbreviation “NK-1” should be given for the first time in the text. Changed

Page 4, line 138: “this tracer…” should be better changed into “11C-MET”. Changed

Page 5, line 210-211: the sentence “A summary of PET radiotracers used in glioma imaging is provided in Table 2” should be changed into “Summary of Current Treatments for Recurrent Glioma is provided in Table 2”. Changed  

Page 5, line 190: (….ratio of 5,36, 7,64 and 6,30) should be changed into (…. ratio of 5,36, 7,64 and 6,30, respectively) Changed

Page 5, last paragraph: The letter size of this paragraph is smaller than the other parts of text. Changed

Page 6, line 221, the sentence “ 1220% and 1398% “ should be changed into “220% and 398%” Changed

Page 8, line 305: the word “my “ should be corrected to “may”. Changed

Page 8, line 130: the full name for the abbreviation “TMZ” should be given for the first time in the text. Changed

The tracers such as 18F-FDG, 11C-MET and 18F-FET should be written in text as 18F-FDG, 11CMET and 18F-FET. The same for other tracers. The mass numbers of tracers should be superscripted. Changed

The References should be carefully checked according to the instructions of authors of “Cancers”. Similar to reviewer 1; we have changed to better fit the requirements

Comments on the Quality of English Language

Only minor editing of English required

Minor revisions for reference numbering; instead of superscript- it is just [#] now. Also reworded highlighted sections that were deemed similar to prior publications and changed for better syntax.

Reviewer 3 Report

Comments and Suggestions for Authors

Review for the manuscript # cancers-2966635-peer-review-v1

entitled “Theranostics In Recurrent Gliomas: A Review” by  Austin R. Hoggarth, Sankar Muthukumar, Steven Thomas, James Crowley, Jackson Kiser, and Mark R Witcher

The manuscript presents a review of the current state of the combined diagnostic imaging and treatment (collectively termed in the paper as theranostics) of high grade gliomas. The topic and title are very interesting and promising, and the manuscript is well written.

I would like to suggest a few critical comments.

First of all, I found it a bit confusing that only radiopharmaceutical approaches are included in the term theranostics. I believe it is much wider. For example, a very broad field of nanotheranostic approaches is not mentioned. I suggest reconsidering the title and make it more specific (e.g., radiopharmaceutical theranostics, or clinical theranostics – I am just thinking aloud about that but a proper brainstorming may help better). Another missing areas, I believe, are fluorescence-guided surgery or hyperspectral imaging of gliomas combined with photodynamic therapy. It is, again, beyond the radiopharmacology.

I believe that the standard-of-care schemes applicable for the high grade gliomas need to be clearly presented. Table 2 summarises on this, however, for instance, it does not mention the specific chemotherapy scheme, and then the advantages, limitations and side effects that are listed are not linked to the specific drug treatment (and therefore it is difficult to comment on whether the summary is correct).

It may be helpful to mention which of the discussed methodologies were approved for the clinical application, and which are at the stages of pre-clinical evaluation and clinical trials.

I have an impression that the references are not in the MDPI format, but please check the rules for the possible updates. Also, the paragraph (rows 205-210) has wrong format.

Finally, I recommend to pass through the paper and carefully adjust the statements where a degree of certainty is not very high and use more specific or milder formulations (e.g., row 56 – should be “damaging to … BBB characteristic for high grade gliomas”; row 88 “theranostic allows” should be “theranostic potentially allows”).

Author Response

Review for the manuscript # cancers-2966635-peer-review-v1 entitled “Theranostics In Recurrent Gliomas: A Review” by Austin R. Hoggarth, Sankar Muthukumar, Steven Thomas, James Crowley, Jackson Kiser, and Mark R Witcher 3

The manuscript presents a review of the current state of the combined diagnostic imaging and treatment (collectively termed in the paper as theranostics) of high grade gliomas. The topic and title are very interesting and promising, and the manuscript is well written.

I would like to suggest a few critical comments.

First of all, I found it a bit confusing that only radiopharmaceutical approaches are included in the term theranostics. I believe it is much wider. For example, a very broad field of nanotheranostic approaches is not mentioned. I suggest reconsidering the title and make it more specific (e.g., radiopharmaceutical theranostics, or clinical theranostics – I am just thinking aloud about that but a proper brainstorming may help better). Another missing areas, I believe, are fluorescence-guided surgery or hyperspectral imaging of gliomas combined with photodynamic therapy. It is, again,beyond the radiopharmacology.

We agree, the field of theranostics is broad. Changed the title to “clinical” to better delineate that. We find that fluorescence-guided surgery, hyperspectral imaging and photodynamics we believe is beyond the scope of this paper.

I believe that the standard-of-care schemes applicable for the high grade gliomas need to be clearly presented. Table 2 summarises on this, however, for instance, it does not mention the specific chemotherapy scheme, and then the advantages, limitations and side effects that are listed are not linked to the specific drug treatment (and therefore it is difficult to comment on whether the summary is correct).

We find that some of this is addressed in the Functional Neuro-Oncologic Implications subsection.

It may be helpful to mention which of the discussed methodologies were approved for the clinical application, and which are at the stages of pre-clinical evaluation and clinical trials.

This is briefly described in Table 1.

I have an impression that the references are not in the MDPI format, but please check the rules for the possible updates. Also, the paragraph (rows 205-210) has wrong format.

Changed

Finally, I recommend to pass through the paper and carefully adjust the statements where a degree of certainty is not very high and use more specific or milder formulations (e.g., row 56 – should be “damaging to … BBB characteristic for high grade gliomas”; row 88 “theranostic allows” should be “theranostic potentially allows”).

Wording changed throughout the manuscript.